# Analysis of the Effect of Carbonation Rate on the Concrete Water Reservoir Structures According to Applied Waterproofing/Anticorrosive Methods

**DOI:** 10.3390/ma15196854

**Published:** 2022-10-02

**Authors:** Jeong-Il Go, Wan-Gu Park, Su-Young Choi, Bo Jiang, Xingyang He, Sang-Keun Oh

**Affiliations:** 1Institute of Biomedical Engineering and Biomaterials, Seoul National University of Science and Technology, 232 Gongneung-ro, Nowon-gu, Seoul 01811, Korea; 2New Material and Laboratory Co., Ltd., 11 Sinchon-ro, Paju-si 10880, Korea; 3School of Civil Engineering and Environment, Hubei University of Technology, No.28, Nanli Road, Hongshan District, Wuchang, Wuhan 430068, China; 4School of Architecture, Seoul National University of Science & Technology, 232 Gongneung-ro, Nowon-gu, Seoul 01811, Korea

**Keywords:** water reservoir structure, water reservoirs, concrete structures, waterproofing method, carbonation rate

## Abstract

This study seeks to analyze how the degree of carbonation and the application of waterproofing and anticorrosive materials affect carbonation in water reservoirs among the water treatment facilities managed by the Seoul Metropolitan Government. To guarantee similarity of the experimental group, 42 highly similar water reservoirs were selected from among the water supply reservoirs currently in operation in Seoul. On-site carbonation assessments were performed in order to derive the carbonation rate coefficients. In the water reservoirs with applied waterproofing and anticorrosive materials immediately after public service, the upper and lower limits were D = 1.13$\sqrt{t}$ and D = 0.29$\sqrt{t}$, respectively, whereas those of the water reservoir applied with waterproofing and anticorrosive materials after 15 years of service life were D = 1.89$\sqrt{t}$ and D = 0.94$\sqrt{t}$, respectively. The comparative analysis showed that the rate of reduction in the carbonation rate was about 10.4% to 16.8% in the water reservoirs applied with waterproofing and anticorrosive methods after 15 years of service life. However, reduction in the carbonation rate was about 46.4% to 74.3% in the water reservoirs applied with waterproofing and anticorrosive methods at the initial stage of service life. It was confirmed that the early application of waterproofing and anticorrosive materials is effective in suppressing carbonation of concrete water reservoir structures.

## 1. Introduction

The majority of water supply and water treatment facility structures (hereafter, water treatment facilities) in South Korea are built using reinforced concrete. Different environmental conditions are created inside the water treatment facility, which is largely divided into the area in air (atmospheric zone) that does not stay in contact with water and the area in water (underwater zone) that is constantly filled with fresh water [1,2]. In the underwater zone, chlorides from chlorine disinfectants used for water purification may contribute to deterioration [3,4,5,6,7,8]. This raises a concern since inside the water treatment facilities that have continuous freshwater environment, chloride penetration into concrete leads to rebar corrosion in the reinforced concrete structure [9,10,11,12]. The atmospheric zone is vulnerable to carbonation. Carbonation refers to a phenomenon where cement hydrates in concrete absorb carbon dioxide and are transformed into calcium carbonate (CaCO_3_), resulting in the loss of alkali properties. In general, carbonation occurs when calcium hydroxide in the pores of the concrete reacts with carbon dioxide (CO_2_) in the air [13,14,15,16,17,18,19]; however, this reaction rarely occurs on dry concrete surfaces because it requires humidity [20]. However, a high humidity environment is created inside the water treatment facility which forms carbonation conditions, thereby posing a high risk of rebar corrosion in concrete. In order to respond to a complex environment that accelerates the deterioration of structures, waterproofing and anticorrosive materials are generally applied to the inside of water treatment facilities, regardless of whether the area is in the underwater zone or atmospheric zone [21,22,23,24]. However, facility managers in the Seoul Waterworks Division, constantly raise concerns that the exposure of CO_2_ gas in the air in the water treatment facility is limited, and that carbonation will be insignificant as a result. However, facility managers in the Seoul Waterworks Division constantly claim that the exposure of CO_2_ gas in the air in the water treatment facility is limited, so carbonation will be insignificant. Given the internal environment of the water treatment facility, this view has no major flaws in theory. However, this view is not based on quantitative data regarding the actual status of carbonation in the field. Therefore, it cannot be a reasonable alternative for determining the validity of applying waterproofing and anticorrosive materials to the water treatment facilities simply based on theory [25,26]. In this regard, studies related to the analysis of the carbonation rate of concrete structures when waterproofing and anticorrosive materials is applied were sought, but none were found.

In this regard, this study aimed to comprehensively analyze the effects of the carbonation degree in the atmospheric zone and the application of waterproofing and anticorrosive materials in water supply reservoirs among the water treatment facilities managed by the Seoul Metropolitan Government. It also sought to provide quantitative data that can be used in determining the validity of applying waterproofing and anticorrosive materials to the atmospheric zone.

## 2. Materials and Methods

### 2.1. Water Supply Reservoir Specifications

To set up an experimental group for this study, water supply reservoirs in operation were selected as targets for evaluation. However, there is a limit in setting up the experimental group with the same conditions, since there are differences in water treatment capacity due to the difference in the amount of water required for each region where the water reservoir is installed. Furthermore, the completion time varies depending on each facility. In order to guarantee the similarity of the experimental group within the possible range, the region was limited to Seoul. A reinforced concrete reservoir constructed with a design strength of 24 MPa was selected from among the water supply reservoirs currently operational in Seoul. In addition, the scope of application was limited to the facilities applied with coating materials, including epoxy series but excluding sheet and panel materials, as shown in Table 1 [27]. 

This study does not seek to compare the degree of carbonation according to the type of waterproofing/anticorrosive material, but to analyze the difference in carbonation rate depending on whether or not a waterproofing/anticorrosive material has been applied. Accordingly, the type of waterproofing/anticorrosive coating material applied to the water supply reservoir selected as the experimental group was not set as a variable. A total of 42 water supply reservoirs that met the above-mentioned conditions were included in the experimental group. The detailed specifications of each water reservoir (as of 2019) are summarized in Table 2.

### 2.2. Assessment of Carbonation Depth

Concrete cores were taken from the atmospheric zone inside the water supply reservoirs in each region corresponding to the selected experimental group. The concrete core was collected from the surface 0.5 m above the maximum water level of the reservoir, as shown in Figure 1. The collected concrete core was then immediately sealed in order to minimize the influence of outside air. The depth of carbonation was primarily assessed at the core sampling part of the wall at the site, as shown in Figure 2a. However, due to dark and poor site conditions, the carbonation depth was remeasured for the collected cores, as shown in Figure 2b in order to ensure the reliability of the assessment. The depth of carbonation was measured within 24 h from the point of collection of the specimens taken from each water reservoir included in the experimental group. The depth of carbonation was measured in accordance with the carbonation depth measurement method specified in KS F 4042:2012. The part that did not turn red when a 1% phenolphthalein solution was applied was measured as the carbonated part [28]. The depth of carbonation was measured using Vernier calipers with the least count of 0.1 mm specified in KS B 5203-1:2013 [29]. 

## 3. Results and Discussion

### 3.1. Carbonation Depth Measurement Results

Table 3 shows the results of carbonation depth measurement based on the concrete cores collected from the actual site for each water supply reservoir.

Based on the result of the carbonation depth measurement for a total of 42 water supply reservoir structures located in Seoul, the carbonation depth was measured to be between 1.2 and 9.8 mm. The sites where waterproofing corrosion protection material was applied from the beginning of the structure’s life did not generally undergo deep carbonation. Compared with the sites with waterproofing/corrosion protection material applied from the beginning of the structure’s life, it was confirmed that carbonation proceeded relatively deeply in the sites where protection was not applied from the beginning.

### 3.2. Carbonation Depth Analysis

To analyze the validity of applying waterproofing and anticorrosive methods, the carbonation depth was analyzed and compared using theoretical equations for carbonation and field measurements. However, in the water reservoir structure, which is the subject of this study, the waterproofing/anticorrosive method was applied from the initial stage of service life, or the method was applied after a certain period of service life. This poses difficulties making direct comparisons with structures not applied with waterproofing/anticorrosive material because there are no sites with no waterproofing/anticorrosive method applied from the initial stage of service life. Based on the structures applied with the waterproofing/anticorrosive method from the initial stage of service life and the structures applied with the waterproofing/anticorrosive method after 15 years of service life (10 to 20 years), the carbonation rate coefficients of the water reservoirs with no waterproofing/anticorrosive method applied from the initial stage of service life were derived in order to conduct a comparative analysis. The carbonation rate coefficient was derived based on the assumption that the carbonation rate (depth) is linearly proportional to the square root of time. The derivation process is as follows. 

#### 3.2.1. Mechanism of Correlation Analysis on Changes in Carbonation Depth Due to the Application of Waterproofing and Anticorrosive Methods

Figure 3 shows the schematic diagram of the correlation between the neutralization rate coefficient and the application of the waterproofing/anticorrosive method. When the waterproofing/anticorrosive method is not applied in the course of the reservoir’s public service, neutralization progresses to a depth of D_2_ in √t_1_ year and D_3_ in √t_2_ year. However, when the waterproofing/anticorrosive method is applied after a certain period of time, the depth of carbonation is D_2_ in √t_1_ year. The neutralization rate coefficient then decreases significantly, showing a depth of D_1_ in √t_2_ year. Therefore, it can be inferred that if the waterproofing/anticorrosive method is applied in √t_1_ year, the carbonation rate coefficient C_3_ with no waterproofing/anticorrosive method applied during service life follows the trend of C_3_, which is the carbonation rate coefficient with the waterproofing/anticorrosive method applied from the initial stage of service life, from √t_1_ year onwards. Accordingly, the identification of the carbonation rate coefficient of the inner wall of the water reservoirs applied with the waterproofing/anticorrosive method in the initial stage of service and that of the water reservoirs applied after a certain period of time makes it possible to derive the carbonation rate coefficient of the water supply reservoirs not applied with the waterproofing/anticorrosive method from the initial stage of service life [30].

#### 3.2.2. Mechanism of Correlation Analysis on Changes in Carbonation Depth Due to the Application of Waterproofing and Anticorrosive Methods

In order to confirm the carbonation rate coefficient of the inner wall of the water reservoirs applied with the waterproofing/anticorrosive method immediately after construction, and that of the water reservoirs applied with the waterproofing/anticorrosive method after a certain period of time, theoretical equations for carbonation [31,32] that consider the environment of the water reservoir were selected in order to derive the carbonation rate coefficients, as shown in Table 4, Table 5 and Table 6. The rate coefficients were represented as the upper and lower limits among the results derived for each water supply reservoir.
Theoretical formula: *x_c_(t) = A_c_*√t, where *A_c_* is the neutralization rate coefficient (mm/√t), and *t* is the yearTheoretical neutralization rate coefficient: *A_c_* = *k*_1_*·α*_1_*·α*_2_*·α*_3_*·β*_1_*·β*_2_*·β*_3_ [33]*k*_1_: Neutralization rate constant 17.2 mm/√t*α*_1_: Coefficient according to the types of aggregates used in concrete*α*_2_: Coefficients according to the types of cement*α*_3_ = *w/c* − 0.38: Coefficient according to water–cement ratio*β*_1_ = 0.017*T* + 0.48*β*_1_: Coefficient according to the temperature, where *T* is the average temperature (°C) of the region*β*_2_: Coefficient according to humidity
Wet environment: *β*_2_ = *k_w_* (*k_w_* is coefficient for wet environment, which is assumed to be 0.2)*β*_3_ =$\sqrt{C_{{\mathrm{Co}}_{2}}∕5.0}$: Coefficient according to carbon dioxide concentration

#### 3.2.3. Comparison between Carbonation Rate Coefficient According to Theoretical Equation and Measured Carbonation Depth

To analyze the validity of the carbonation rate coefficient derived according to the theoretical equation, the carbonation rate coefficient was compared with the carbonation depth measured at the site. As shown in Figure 4 and Figure 5, the data on the carbonation depth obtained from on-site measurements are displayed on graphs that show the carbonation depth distributions in the water supply reservoirs with the waterproofing/anticorrosive method applied from the initial stage of service life and after 15 years of service life, respectively. The graphs of the carbonation rate coefficients derived using the theoretical equation are overlaid for comparison.

As shown in Figure 4 and Figure 5, the scatter graphs showing the degree of carbonation measured at the site were overlaid with the graphs that show the carbonation rate coefficients according to the theoretical equation. The comparison revealed that the range of values was similar to each other, thereby confirming the reliability of the data. However, there was a measured carbonation depth value that was out of the range of the carbonation rate coefficient derived by using the theoretical equation. In this study, a research analysis will be conducted based on the carbonation values measured at the site of the water reservoir structures. Therefore, it was necessary to adjust the range of the carbonation rate coefficient according to the theoretical equation to include the range for the measured carbonation depth.

#### 3.2.4. Adjustment of Carbonation Rate Coefficient Range According to the Theoretical Equation

The range of the carbonation rate coefficient according to the theoretical equation was adjusted in order to include all the values for the carbonation depth measured at the site, as shown in Figure 4 and Figure 5. The adjustment results are shown in Table 7 and Figure 6 and Figure 7.

#### 3.2.5. Derivation and Comparison of Carbonation Rate Coefficients of Water Reservoirs with No Waterproofing/Anticorrosive Method Applied after Public Service

The derivation of the carbonation rate coefficient of the water reservoirs with no waterproofing/anticorrosive method applied after public use follows the steps shown in Figure 8. In order to derive the carbonation depth at √t_1_, the duration of the waterproofing/anticorrosive method application to the water reservoirs after a certain period of time during public service, a graph of the carbonation rate (C_1_) of the water reservoirs with the waterproofing/anticorrosive method applied at the initial stage of public service, and a graph of the carbonation rate (C_2_) of the water reservoirs with the waterproofing/anticorrosive method applied after a certain period of time during public service were drawn to derive D_1_ at √t_2_. Because the carbonation depth of the water reservoirs applied with the waterproofing/anticorrosive method after a certain period of time during public service in sections √t_1_ to √t_2_ follows the trend of the carbonation rate (C_1_) of the water reservoirs applied with the waterproofing/anticorrosive method at the initial stage of public use, D_2_, the carbonation depth at √t_1_, the point of time for the application of the waterproofing/anticorrosive method was derived by moving the graph of the carbonation rate (C_1_) of the water reservoirs with the waterproofing/anticorrosive method applied at the initial stage of public use to be parallel to D_1_. Based on D_2_, the graph of the carbonation rate (C_3_) of the water reservoirs with no waterproofing/anticorrosive method applied after public use was drawn and extended to the point
$\sqrt{t}$_2_. The carbonation rates (C_1_, C_2_, and C_3_) for each point in time for the waterproofing/anticorrosive method application were subsequently compared with one another.

The adjusted carbonation rate coefficient of the water reservoirs applied with the waterproofing/anticorrosive method immediately after public service was derived from Section 3.2.4. The adjusted carbonation rate coefficient of the water reservoirs applied with the waterproofing/anticorrosive method after 15 years of service life and that of the water reservoirs with no waterproofing/anticorrosive method applied after public service were derived by using the method shown in Figure 8. Based on these results, the carbonation rate coefficient of the water reservoirs with no waterproofing/anticorrosive method applied after public service was derived, as shown in Figure 9 and Figure 10. The carbonation rate coefficients according to the duration of the derived waterproofing/anticorrosive method application are summarized in Table 8.

The carbonation rate coefficients were derived according to the duration of the waterproofing/anticorrosive method application. The carbonation rate coefficient of the water reservoirs with the waterproofing/anticorrosive method applied immediately after public service and that of the water reservoirs applied with the waterproofing/anticorrosive method after a certain period of time during public service were compared, as shown in Figure 11 and Figure 12. The comparison results show that when the waterproofing/anticorrosive method was applied from the initial stage of public service, the carbonation rate was reduced when compared with that of the water reservoirs with no waterproofing/anticorrosive method applied after public service.

Figure 13 compares the rates of decrease in carbonation rates with or without application of the waterproofing/anticorrosive method based on the carbonation rate coefficients and duration of the waterproofing/anticorrosive method application. The comparison reveals that the rate of decrease in the carbonation rate was about 10.4% to 16.8% in the water reservoirs with the waterproofing/anticorrosive method applied after 15 years of service life, whereas the decrease in the carbonation rate was about 46.4% to 74.3% in the water reservoirs applied with the waterproofing/anticorrosive method from the initial stage of public service. This result suggests that the application of the waterproofing/anticorrosive method is needed in order to decrease the carbonation rate of the water reservoir structures.

## 4. Conclusions

This study investigated the carbonation status of concrete water reservoir structures installed underground. Comprehensive analysis of the effects of waterproofing and anticorrosive materials on carbonation was carried out and the results are as follows:
(1)To guarantee the similarity of the experimental group, 42 highly similar water supply reservoirs were selected from among the water reservoirs that are currently in operation in Seoul. On-site carbonation assessments were then performed in order to derive the carbonation rate coefficients. In the case of the water reservoirs applied with the waterproofing/anticorrosive method immediately after public service, the upper and lower values were D = 1.13$\sqrt{t}$ and D = 0.29$\sqrt{t}$, respectively, whereas those for the water reservoirs with the waterproofing/anticorrosive method applied after 15 years of service life were D = 1.89$\sqrt{t}$ and D = 0.94$\sqrt{t}$, respectively.(2)The carbonation rate coefficient of the water reservoirs with no waterproofing/anticorrosive method applied from the initial stage of public service was derived based on the carbonation rate coefficients of the structures applied with the waterproofing/anticorrosive method from the initial stage of public service and after 15 years of service life. The results show that for the carbonation rate coefficient of the water reservoirs with no waterproofing/anticorrosive method applied from the initial stage of public service, the upper and lower limits were D = 2.11$\sqrt{t}$ and D = 1.13$\sqrt{t}$, respectively.(3)The carbonation rates of the water reservoirs with or without application of the waterproofing/anticorrosive method were analyzed based on the carbonation rate coefficients and duration of the waterproofing/anticorrosive method application derived above. Results show that the carbonation rate of the water reservoirs with the waterproofing/anticorrosive method applied from the initial stage of public service or during public service was reduced when compared with that of the water reservoirs with no waterproofing/anticorrosive method applied after public service.(4)The rate of decrease in the carbonation rate was about 10.4% to 16.8% in the water reservoirs applied with the waterproofing/anticorrosive method after 15 years of service life. The decrease in carbonation rate was about 46.4% to 74.3% in the water reservoirs applied with the waterproofing/anticorrosive method from the initial stage of public service.(5)Based on the above research results, it is concluded that the application of the waterproofing/anticorrosive method is needed in order to decrease the carbonation rate. Early application is thus recommended for a concrete water reservoir inside the water reservoir structure.

The carbonation rate coefficient analysis performed in this study did not consider various influencing factors on carbonation due to the limitations on field conditions. Consequently, the correlation coefficient of the distribution is low since the main factors are selected and analyzed. In order to overcome this problem, additional analysis applying detailed environmental conditions (CO_2_ concentration in the atmosphere, temperature, humidity, etc.) should be performed. 

However, this study’s contribution is anchored on its quantitative analysis and presentation of the effect of applying the waterproofing and anticorrosive method inside concrete water reservoir structures on suppressing carbonation. The results of this study are also expected to be used as basic data for rational decision making when applying waterproofing and rust prevention methods to reservoir structures in the future.

## Figures and Tables

**Figure 1 materials-15-06854-f001:**
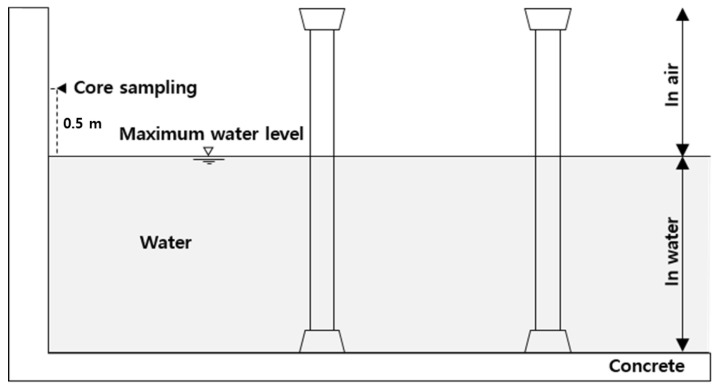
Conceptual diagram of core sampling location in the water reservoir.

**Figure 2 materials-15-06854-f002:**
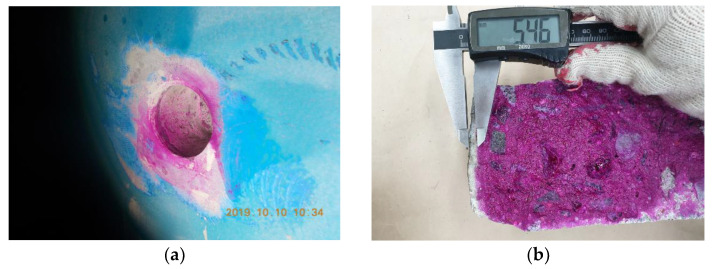
Carbonation depth measurement. (**a**) Measurement of carbonation depth in the core sampling area. (**b**) Measurement of carbonation depth of the collected sample core.

**Figure 3 materials-15-06854-f003:**
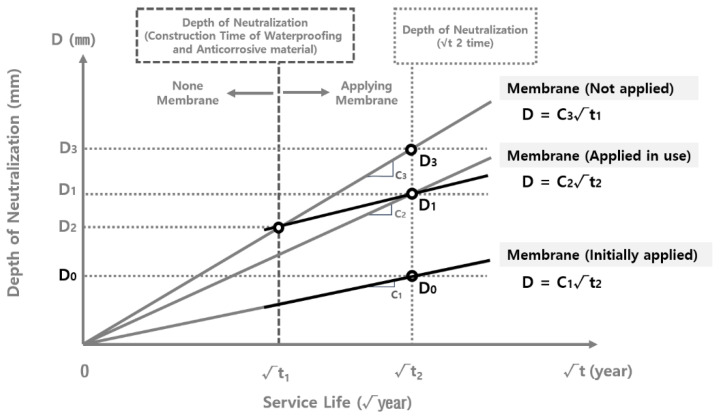
Schematic diagram of correlation between the carbonation rate coefficient and the application of the waterproofing/anticorrosive method.

**Figure 4 materials-15-06854-f004:**
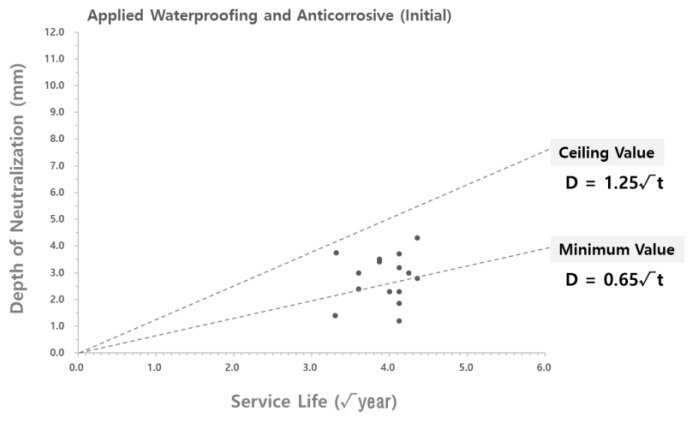
Carbonation depth distribution on the inner wall of the water reservoirs (waterproofing/anticorrosive method applied from the initial stage of service life) - according to the theoretical equation.

**Figure 5 materials-15-06854-f005:**
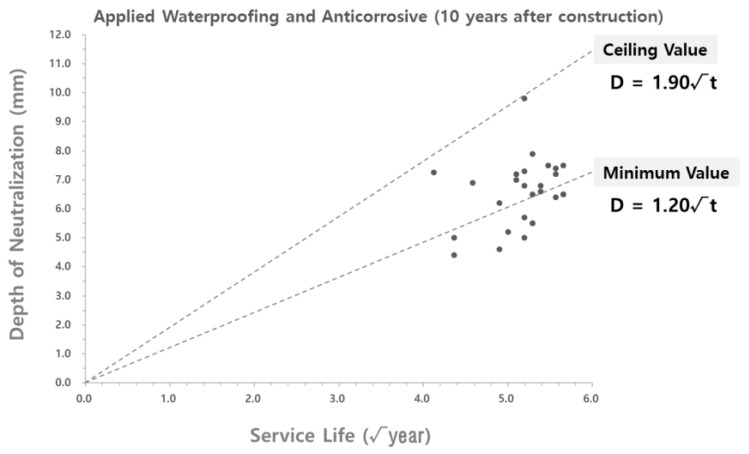
Carbonation depth distribution on the inner wall of the water reservoirs (waterproofing/anticorrosive method applied after 10 years of service life) - according to the theoretical equation.

**Figure 6 materials-15-06854-f006:**
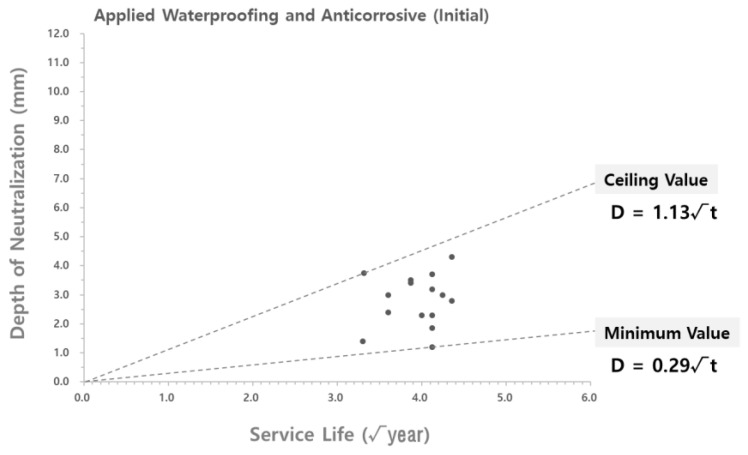
Carbonation depth distribution on the inner wall of the water reservoirs (waterproofing/anticorrosive method applied from the initial stage of service life) - reflect on-site adjustments.

**Figure 7 materials-15-06854-f007:**
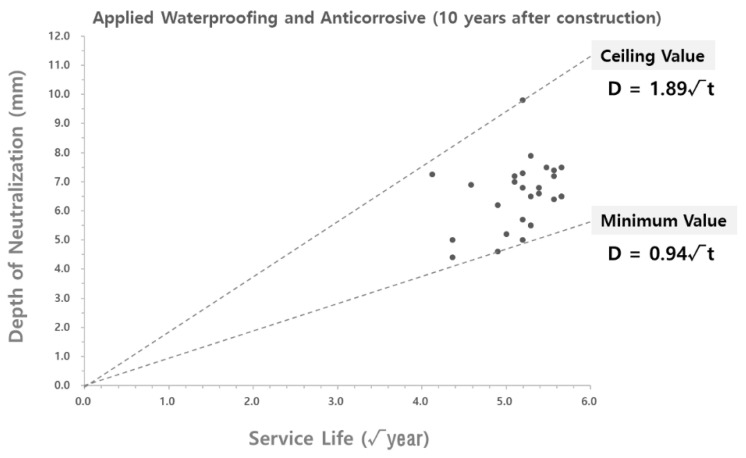
Carbonation depth distribution on the inner wall of the water reservoirs (waterproofing/anticorrosive method applied after 10 years of service life) - reflect on-site adjustments.

**Figure 8 materials-15-06854-f008:**
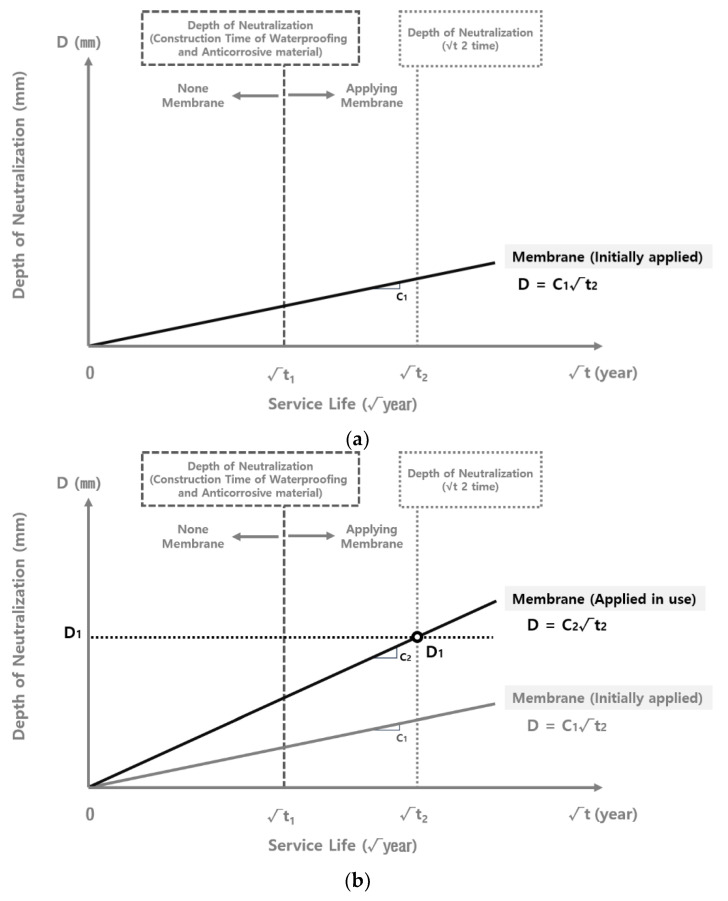
Carbonation depth distribution at the inner wall of water reservoirs (waterproofing/anticorrosive method applied after 15 years of service life): (**a**) Carbonation depth at the initial stage of application; (**b**) Derivation of D_1_ with a graph of application during public service; (**c**) Derivation of D_2_ with parallel translation of a graph of application at the initial stage of public service; (**d**) Derivation of C_3_ at √t_1_ with D_2_; (**e**) Derivation of D_3_ at √t_2_ through the extension of the derived line; (**f**) Comparison according to the duration of the waterproofing/anticorrosive method application.

**Figure 9 materials-15-06854-f009:**
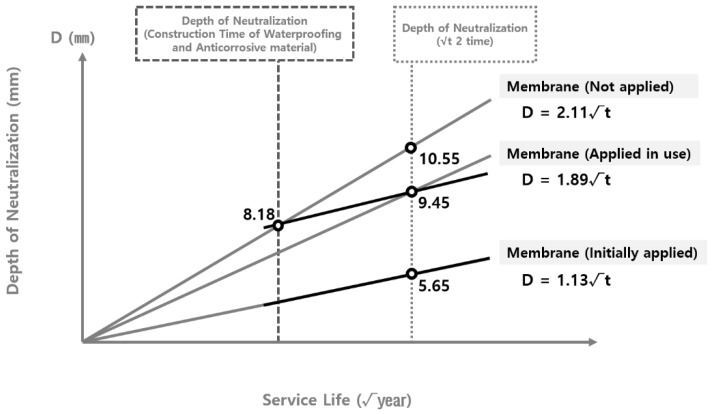
Derivation of carbonation rate coefficients of the water reservoirs with no waterproofing/anticorrosive method applied (upper limits).

**Figure 10 materials-15-06854-f010:**
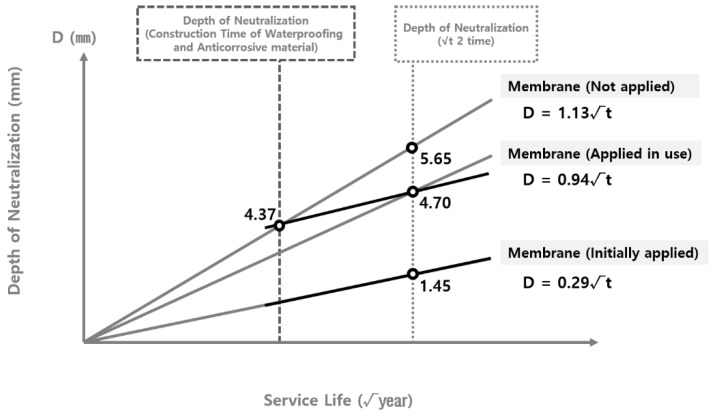
Derivation of carbonation rate coefficients of the water reservoirs with no waterproofing/anticorrosive method applied (lower limits).

**Figure 11 materials-15-06854-f011:**
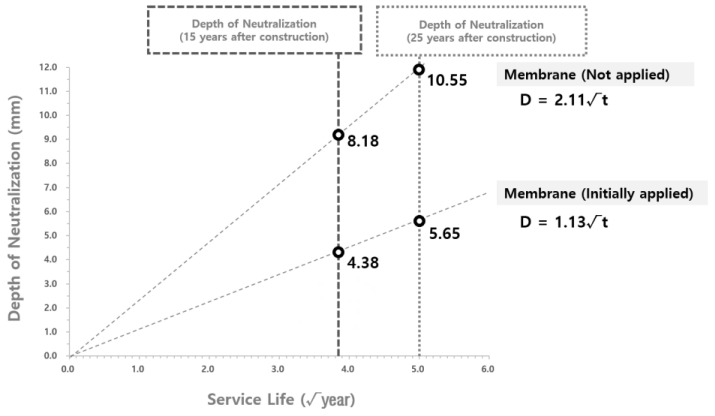
Comparison of carbonation rate coefficients of water reservoirs with or without application of the waterproofing/anticorrosive method (upper limits).

**Figure 12 materials-15-06854-f012:**
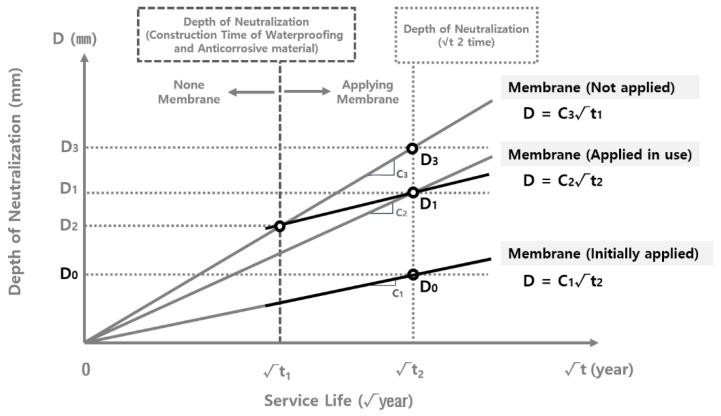
Comparison of carbonation rate coefficients of water reservoirs with or without application of the waterproofing/anticorrosive method (lower limits).

**Figure 13 materials-15-06854-f013:**
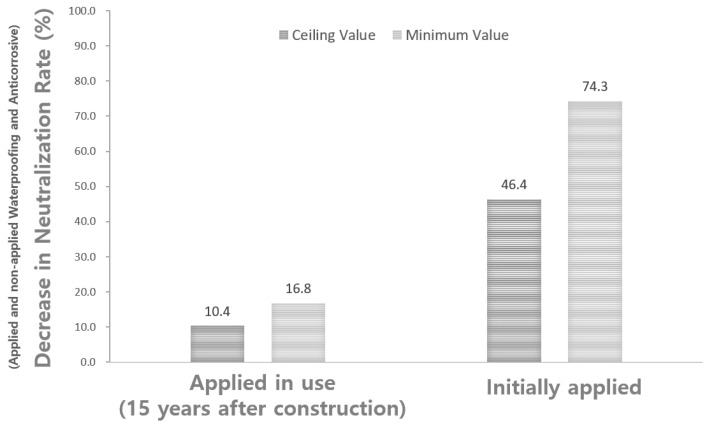
Comparison of rates of decrease in carbonation rates according to the duration of the waterproofing/anticorrosive method application.

**Table 1 materials-15-06854-t001:** Characteristic of coating materials applied to concrete water reservoir structures.

Coating Materials	Constituent	CAS NO.	Content (%)
Epoxy	Diglycidyl ether of bispheonl F	2095-03-6	65
	Limestone	1317-65-3	30
	etc.	-	5
Ceramic Metal	Diglycidyl ether of bispheonl F	2095-03-6	40
	Micro ceramic metal powder	-	20
	etc.	-	40
Polyurea	4,4-Diphenylmethane diisocyanate	101-68-8	55
	1,6-Hexamethylene diisocyanate	28182-81-2	40
	etc.	-	5

**Table 2 materials-15-06854-t002:** Specifications for each water supply reservoir.

No.	Completion (Year)	Design Strength (MPa)	Service Life (Year)	Capacity (m^3^/Day)	Compressive Strength (MPa)	Duration of Waterproofing/Anticorrosive Material Application (Year)
1	2003	24	16	2500	27	16
2	2002	24	17	50,000	28.6	17
3	1987	21	32	60,000	26.1	17
4	1993	24	26	600	28.5	10
5	2006	24	13	60,000	24.3	13
6	1987	21	32	23,000	21.3	16
7	1990	24	29	40,000	26	14
8	1987	24	32	30,000	27.2	14
9	2000	24	19	60,000	24.8	8
10	1990	24	29	60,000	24.5	12
11	1992	24	27	2500	21.9	8
12	2001	24	18	110,000	28.1	18
13	1995	24	24	60,000	29.1	9
14	1989	24	30	2500	25.9	13
15	2004	30	15	30,000	31.2	15
16	2004	30	15	20,000	30	15
17	1988	24	31	2000	24.7	12
18	2008	27	11	10,000	28.5	11
19	2002	27	17	200,000	30	17
20	2014	30	5	3000	32.9	5
21	1993	24	26	100,000	28.1	12
22	2002	24	17	40,000	26	17
23	1998	21	21	170,000	27.3	8
24	1988	21	31	2000	23.1	12
25	1991	21	28	1200	23.4	9
26	1992	21	27	3000	26.7	9
27	2002	24	17	200,000	26.4	17
28	1995	21	24	6000	24.8	8
29	1992	24	27	4000	25.8	12
30	1985	21	19	11,000	23.6	8
31	2000	21	19	145,000	25.1	19
32	2002	24	17	90,000	24.8	17
33	2000	24	19	40,000	23.9	19
34	1991	24	28	4000	25.1	11
35	1991	24	28	60,000	24.2	10
36	1987	21	31	200	24	19
37	1991	21	28	3300	24.1	13
38	2006	27	13	20,000	26.8	13
39	1994	21	25	2000	26.1	12
40	2002	24	17	10,000	23.6	3
41	1992	24	27	2000	25.8	12
42	1992	24	27	1000	25.3	12

**Table 3 materials-15-06854-t003:** Carbonation depth measurement results for each water supply reservoir.

No.	Completion (Year)	Design Strength (MPa)	Service Life (Year)	Capacity (m^3^/Day)	Compressive Strength (MPa)	Carbonation Depth (mm)	Duration of Waterproofing/Anticorrosive Material Application (Year)
1	2003	24	16	2500	27	2.3	16
2	2002	24	17	50,000	28.6	3.2	17
3	1987	21	32	60,000	26.1	6.5	17
4	1993	24	26	600	28.5	7.2	10
5	2006	24	13	60,000	24.3	2.4	13
6	1987	21	32	23,000	21.3	7.5	16
7	1990	24	29	40,000	26	6.8	14
8	1987	24	32	30,000	27.2	6.5	14
9	2000	24	19	60,000	24.8	5.0	8
10	1990	24	29	60,000	24.5	6.6	12
11	1992	24	27	2500	21.9	9.8	8
12	2001	24	18	110,000	28.1	3.0	18
13	1995	24	24	60,000	29.1	6.2	9
14	1989	24	30	2500	25.9	7.5	13
15	2004	30	15	30,000	31.2	3.4	15
16	2004	30	15	20,000	30	3.5	15
17	1988	24	31	2000	24.7	7.2	12
18	2008	27	11	10,000	28.5	3.8	11
19	2002	27	17	200,000	30	1.2	17
20	2014	30	5	3000	32.9	1.4	5
21	1993	24	26	100,000	28.1	7.0	12
22	2002	24	17	40,000	26	3.7	17
23	1998	21	21	170,000	27.3	6.9	8
24	1988	21	31	2000	23.1	7.4	12
25	1991	21	28	1200	23.4	5.5	9
26	1992	21	27	3000	26.7	5.0	9
27	2002	24	17	200,000	26.4	1.9	17
28	1995	21	24	6000	24.8	4.6	8
29	1992	24	27	4000	25.8	5.7	12
30	1985	21	19	11,000	23.6	4.4	8
31	2000	21	19	145,000	25.1	2.3	19
32	2002	24	17	90,000	24.8	2.7	17
33	2000	24	19	40,000	23.9	4.3	19
34	1991	24	28	4000	25.1	6.5	11
35	1991	24	28	60,000	24.2	7.9	10
36	1987	21	31	200	24	6.4	19
37	1991	21	28	3300	24.1	5.5	13
38	2006	27	13	20,000	26.8	3.0	13
39	1994	21	25	2000	26.1	5.2	12
40	2002	24	17	10,000	23.6	7.3	3
41	1992	24	27	2000	25.8	6.8	12
42	1992	24	27	1000	25.3	7.3	12

**Table 4 materials-15-06854-t004:** Coefficients depending on the types of concrete according to aggregate types.

Item	Types of Concrete According to the Aggregate Type		
	Normal Concrete	Lightweight Aggregate Concrete Class 1 a	Lightweight Aggregate Concrete Class 2 a
*α* _1_	1.0	1.2	1.4

**Table 5 materials-15-06854-t005:** Coefficients according to the types of cement.

Item	Types of Cement					
	Ordinary Portland Cement	High Early Strength Portland Cement	Blast Furnace Cement Class A	Blast Furnace Cement Class B	Blast Furnace Cement Class C	Flay Ash Cement Class B
*α* _2_	1.0	0.85	1.25	1.4	1.8	1.8

**Table 6 materials-15-06854-t006:** Results of carbonation rate coefficients of concrete inner walls in the water reservoirs.

Classification	Time of Waterproofing/Anticorrosive Method Application	Carbonation Rate Coefficient
Membrane coatings	Applied at the initial stage of public service	∘ Upper limit: D = 1.25$\sqrt{t}$ ∘ Lower limit: D = 0.65$\sqrt{t}$
	Applied after 15 years of service life	∘ Upper limit: D = 1.90$\sqrt{t}$ ∘ Lower limit: D = 1.20$\sqrt{t}$

**Table 7 materials-15-06854-t007:** Results of carbonation rate coefficients of concrete inner walls in the water reservoirs.

Classification	Duration of Waterproofing/Anticorrosive Method Application	Carbonation Rate Coefficient
Membrane coatings	Applied at the initial stage of public service	∘ Upper limit: D = 1.13$\sqrt{t}$ ∘ Lower limit: D = 0.29$\sqrt{t}$
	Applied after 15 years of service life	∘ Upper limit: D = 1.89$\sqrt{t}$ ∘ Lower limit: D = 0.94$\sqrt{t}$

**Table 8 materials-15-06854-t008:** Carbonation rate coefficients according to the duration of waterproofing/anticorrosive method application.

Classification	Duration of Waterproofing/Anticorrosive Method Application	Carbonation Rate Coefficient
Membrane coatings	Applied at the initial stage of public service	∘ Upper limit: D = 1.13$\sqrt{t}$ ∘ Lower limit: D = 0.29$\sqrt{t}$
	Applied after 15 years of service life	∘ Upper limit: D = 1.89$\sqrt{t}$ ∘ Lower limit: D = 0.94$\sqrt{t}$
	Not applied after public service	∘ Upper limit: D = 2.11$\sqrt{t}$ ∘ Lower limit: D = 1.13$\sqrt{t}$

## Data Availability

Not applicable.

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
