# Peer review of "Analysis of the Effect of Carbonation Rate on the Concrete Water Reservoir Structures According to Applied Waterproofing/Anticorrosive Methods"

_materials, 2022, doi:10.3390/ma15196854_

Round 1
Reviewer 1 Report
1. Shorten the abstract, It is too large.
2. Please add more discussion in literature review section. Add some recent papers to strengthen the literature. The below paper can help you to strengthen:
Development of hybrid models using metaheuristic optimization techniques to predict the carbonation depth of fly ash concrete
3. The author must define the carbonation of structure first and should the novelty of the work.
4. The paper is poorly written, its need to be improved significantly.
5. It is suggested to divide the conclusions based on the significant results obtained from the study. And limitation of the work should be mentioned.
Author Response
Thanks for the review on the paper.
Comments and Suggestions have been faithfully modified as in the attached file.
We thank you for making this paper more faithful and strengthened through the review.
We will try our best to write a more advanced thesis in the future by referring to the review.
Thank you again.

Reviewer 2 Report
This study attempts to evaluate the effect of carbonation rate on actual concrete structures by utilizing waterproofing agents. the topic of the study is very interesting and of great significance since it provides information on the actual on-site performance of the structure over a long period of time. Nonetheless, the present study in its current form requires some revisions, which are suggested as following:
Abstract:
1. seeks to - not sought
2. which quantitative data? and why? (line 20)
3. what are the mentioned coefficients? why are they even important?
4. Lines 28-33 is a single sentence and not understandable. Please revise.
The reviewer suggests that the authors overhaul the abstract in shorter and more coherent form.
Introduction:
1. Line 60, it is better to use the term 'humidity / moisture' rather than 'water'. Authors can see 'the effect of curing regime on physico-mechanical, microstructural and durability properties of alkali-activated materials: a review' that discusses this phenomenon for alkali-activated materials which are more prone to this issue.
2. Please provide reference for lines 67-68.
3. the authors can also discuss the impact of water on other durability properties (only a suggestion).
4. The introduction section needs a literature review. In other words, the authors should provide a background research analysis to also differentiate this study with those of conducted before.
Materials and methods section
1. This section is well-written but it would have been great if the authors could provide more information on the type of actual materials used. for instance, line 95 says, epoxy is used as a coating material. As the authors might know there are different types of epoxies which type has been used? you can refer to 'epoxy, polyester and vinyl ester based polymer concrete a review to see the different types of epoxies usually available and used.
2. If possible, please cite the codes that is noted in the study (e.g., KS B 5203-1:2013)
Results:
1. If possible please add more explanation to section 3.1.
2. similar to the introduction, the results section also lacks referring to previous studies. for instance, the carbonation rate coefficients outlined can be compared to other studies and their respective findings.
Conclusion:
If possible, the reviewer suggests that the authors write the conclusion in point-by-point basis.
Author Response
Thanks for the Comments and Suggestions on this paper.
We have sincerely corrected the content.
Modifications are as in the attached file.
We thank you for strengthening the thesis through the review.
I will try my best to write a better thesis by referring to the review you suggested when writing a new thesis in the future.
Thank you again.

Reviewer 3 Report
Review manuscript entitled: " Analysis of the effect of Carbonation Rate on the Concrete Water Reservoir Structure according to Applied Waterproofing/Anticorrosive Methods"
The manuscript deals with how the degree of carbonation and application of waterproofing and anticorrosive materials affect carbonation in water reservoirs among the water treatment facilities managed by the Seoul Metropolitan Government. The paper uses a clear scientific approach to the subject matter, which is a clear strength.
However, improvements are required in certain important aspects of the paper, along with some minor improvements:
- Lines 30-32 “Based on the above research results, it is concluded that needed to early application of waterproofing and anticorrosive materials for decrease the carbonation rate.” Please rephrase
- Lines 468-473 “This study comprehensively analyzed the effects of the application of waterproofing and anticorrosive materials on carbonation in inside the underground concrete water reservoir structure among the water treatment facilities managed by the Seoul Metropolitan Government. Based on the quantitative data, the analysis results on current status of carbonation and needs to waterproofing/anticorrosive methods are as follows:” Please rephrase
- Line 540: replace “In this study, he carbonation rate” with “In this study, the carbonation rate”
- Line 546: “However, this study is meaningful in that it compared and analyzed the degree of carbonation according to the application of waterproofing and rust prevention methods for concrete water reservoir structure in use.” Please rephrase
Final Suggestion:
The concept of the manuscript is good, and the scientific approach is solid.
However, extensive editing of the English language is needed, since the manuscript contains many grammatical and syntactic errors.
Author Response
Thanks for the Comments and Suggestions on this paper.
Modifications are as in the attached file.
Once again, thank you for your review on this paper.

Round 2
Reviewer 1 Report
The paper is accepted as it is. To improve the quality of paper the author may cite the following paper:
10.1007/s41062-021-00714-7
10.1007/s41062-020-00447
Author Response
Thanks for more review on the paper.
we modified the article to improve the quality.
Thank you.
